# Strain control of a bandwidth-driven spin reorientation in Ca₃Ru₂O₇

C. D. Dashwood [1,10] ✉, A. H. Walker[1,10] ✉, M. P. Kwasigroch[2,3], L. S. I. Veiga[1,4], Q. Faure[1,5], J. G. Vale[1], D. G. Porter [4], P. Manuel[6], D. D. Khalyavin[6], F. Orlandi [6], C. V. Colin [7], O. Fabelo [8], F. Krüger[1,6], R. S. Perry [1], R. D. Johnson[9], A. G. Green [1] & D. F. McMorrow [1]

The layered-ruthenate family of materials possess an intricate interplay of structural, electronic and magnetic degrees of freedom that yields a plethora of delicately balanced ground states. This is exemplified by Ca₃Ru₂O₇, which hosts a coupled transition in which the lattice parameters jump, the Fermi surface partially gaps and the spins undergo a 90° in-plane reorientation. Here, we show how the transition is driven by a lattice strain that tunes the electronic bandwidth. We apply uniaxial stress to single crystals of Ca₃Ru₂O₇, using neutron and resonant x-ray scattering to simultaneously probe the structural and magnetic responses. These measurements demonstrate that the transition can be driven by externally induced strain, stimulating the development of a theoretical model in which an internal strain is generated self-consistently to lower the electronic energy. We understand the strain to act by modifying tilts and rotations of the RuO₆ octahedra, which directly influences the nearest-neighbour hopping. Our results offer a blueprint for uncovering the driving force behind coupled phase transitions, as well as a route to controlling them.

The coupling between structural and electronic degrees of freedom in quantum materials generates a variety of ground states and drives transitions between them. Textbook examples include the Peierls transition in 1D materials, in which a periodic lattice deformation leads to a metal-insulator transition and the formation of a charge-density wave[1,2]. Similarly, the cooperative Jahn-Teller effect describes the spontaneous distortion of a crystalline lattice to lower the electronic degeneracy and give rise to orbital ordering[3]. Recently, there has been considerable interest in twisted bilayer materials, which host a spectrum of electronic phases—from Mott insulators[4] to unconventional superconductors[5] as the bandwidth is tuned by the twist angle. The key role of the lattice in many quantum materials offers a powerful set of control parameters with which to tune their phases, but also presents a considerable challenge in developing a comprehensive understanding of the interactions that give rise to these phases.

In this context, the application of stress has arisen as a powerful method to tune the electronic properties of quantum materials, including superconducting[6,7], charge/spin-density wave[8–12], nematic[13–16] and topological[17,18] phases. Here, we use uniaxial stress to drive a coupled spin reorientation and Fermi surface reconstruction in Ca₃Ru₂O₇, uncovering the central role of the lattice and facilitating a microscopic understanding of the transition.

Ca₃Ru₂O₇ is a bilayer member of the Ruddlesden-Popper ruthenates, A_{n+1}Ru_nO_{3n+1}, across which varying structural distortions lead to

¹London Centre for Nanotechnology and Department of Physics and Astronomy, University College London, London WC1E 6BT, UK. ²Department of Mathematics, University College London, London WC1H 0AY, UK. ³Trinity College, Cambridge CB2 1TQ, UK. ⁴Diamond Light Source, Harwell Science and Innovation Campus, Didcot, Oxfordshire OX11 0DE, UK. ⁵Laboratoire Léon Brillouin, CEA, CNRS, Université Paris-Saclay, CEA-Saclay 91191 Gif-sur-Yvette, France. ⁶ISIS Neutron and Muon Source, STFC Rutherford Appleton Laboratory, Didcot, Oxfordshire OX11 0QX, UK. ⁷Université Grenoble Alpes, CNRS, Institut Néel, 38000 Grenoble, France. ⁸Institut Laue-Langevin, 71 Avenue des Martyrs, CS 20156, 38042 Grenoble, France. ⁹Department of Physics and Astronomy, University College London, London WC1E 6BT, UK. ¹⁰These authors contributed equally: C. D. Dashwood, A. H. Walker. ✉e-mail: cameron.dashwood.17@ucl.ac.uk; a.walker.17@ucl.ac.uk

a diversity of ground states. In the monolayer compounds these range from superconductivity in $Sr_2RuO_4$[6,19] to a Mott insulating state in $Ca_2RuO_4$[20,21], while bilayer $Sr_3Ru_2O_7$ displays an electronic nematic phase with spin-density wave order[22–25]. $Ca_3Ru_2O_7$ crystallises in the $Bb2_1m$ space group ($a \approx 5.3$ Å, $b \approx 5.5$ Å, $c \approx 19.5$ Å), in which a combination of octahedral tilts around **b** ($X_3^-$, $a^-a^-c^0$ in Glazer notation) and rotations around **c** ($X_2^+$, $a^0a^0c^+$) combine to unlock polar lattice displacements[26]. It undergoes a coupled structural, electronic and magnetic transition, where the lack of inversion symmetry causes a magnetic cycloid to form and mediate a spin-reorientation transition (SRT). Below $T_N \approx 60$ K, the spins align along **a**, coupled ferromagnetically within the bilayers and antiferromagnetically between them, in the $AFM_a$ phase (see Fig. 1b)[27]. On cooling through 49 K, an incommensurate cycloid (ICC) develops with $\mathbf{q} = (\delta, 0, 1)$ ($\delta \approx 0.023$) and the spins rotating in the **a**–**b** plane[28,29]. The envelope of the cycloid evolves from elongated along **a**, to circular, to elongated along **b**. The SRT concludes at 46 K with the cycloid collapsing into the collinear $AFM_b$ phase, where the spins are globally rotated by 90° from $AFM_a$ (Fig. 1d). The SRT is accompanied by a rapid change in the lattice parameters[30] and a partial gapping of the Fermi surface[31–34].

Although there have been various proposals for the mechanism behind the SRT, including recent work that attributes it to the energy gain associated with a Rashba-based hybridisation of bands at the Fermi level[34], these proposals have generally neglected the structural degrees of freedom. We use piezoelectric cells to apply continuously tuneable uniaxial stresses to $Ca_3Ru_2O_7$ single crystals. A combination of neutron and resonant x-ray scattering enables us to directly probe the magnetic structure and offer insight into the response of the lattice to applied stress. Our measurements reveal that the SRT can be fully driven by strain at fixed temperature, and allow the construction of temperature-strain phase diagrams for orthogonal in-plane stresses. This motivates the development of a minimal theoretical model in which strain tunes the electronic hopping. As well as reproducing our strain data, a self-consistent solution of the model (with realistic values for the electronic parameters) reveals that the transition with temperature is driven by an internal strain that arises to lower the electronic energy. We interpret this strain as acting via the tilts and rotations of the $RuO_6$ octahedra, as corroborated by temperature-dependent diffraction measurements. Our results therefore offer a promising mechanism by which to drive electronic and magnetic phase transitions in correlated perovskite materials.

## Results

### Driving the spin reorientation with strain

As a well-established bulk probe of magnetic order, neutron scattering is a natural choice for our experiments. The large sample size generally needed is, however, at odds with the inverse scaling of achievable strain on sample length – an issue that is compounded by the increased background scattering from the strain cell. To overcome these issues, we used the latest generation of piezoelectric strain cells from Razorbill Instruments[35], which can apply large strains to samples measured in a transmission scattering geometry with minimal background from the cell. Combined with the high count-rate of the WISH instrument at the ISIS Neutron and Muon Source[36], this enabled us to achieve sufficient signal at strains of up to 0.5%. A schematic of the strain cell is shown in Fig. 1a, with a $Ca_3Ru_2O_7$ sample spanning a distance $L$ between the sample plates. Applying a voltage across the piezoelectric stacks changes the distance between the plates by $\Delta L$, producing an applied strain $\Delta L/L$ along the $a$-axis of the sample. Further experimental details can be found in the Methods.

Figure 1c shows the result of a strain sweep at a fixed temperature in the centre of the ICC phase. We tracked the intensities of the commensurate $(0, 0, 1)$ magnetic peak, which arises from both the $AFM_a$ and $AFM_b$ structures, and satellite $(\pm\delta, 0, 1)$ peaks that arise from the mediating cycloidal phase. At $\Delta L/L = 0$, we see intense satellite peaks and a small remnant commensurate peak due to temperature and strain gradients in the sample. Under both tensile ($\Delta L/L > 0$) and compressive ($\Delta L/L < 0$) strain the satellite peaks are suppressed and the commensurate peak is simultaneously enhanced. The commensurate intensity is lower under compression, consistent with entering the $AFM_a$ phase, while the higher commensurate intensity under tension is consistent with the $AFM_b$ phase[28]. This assignment is confirmed by strain dependences at temperatures just above and below the SRT (see the Supplementary Information). The slight change in incommensurate intensity at low strains is consistent with the variation of moment size through the ICC phase. Our neutron data therefore provide strong evidence that we can continuously drive the cycloid-mediated SRT with strain at fixed temperature.

Despite this, our neutron setup has a number of drawbacks. The transitions between the magnetic phases can be seen to occur over a finite width of around 0.3% due to the neutron beam illuminating a large region of the sample across which there is a significant strain gradient (see Fig. 1a). Further, despite the large displacements that can

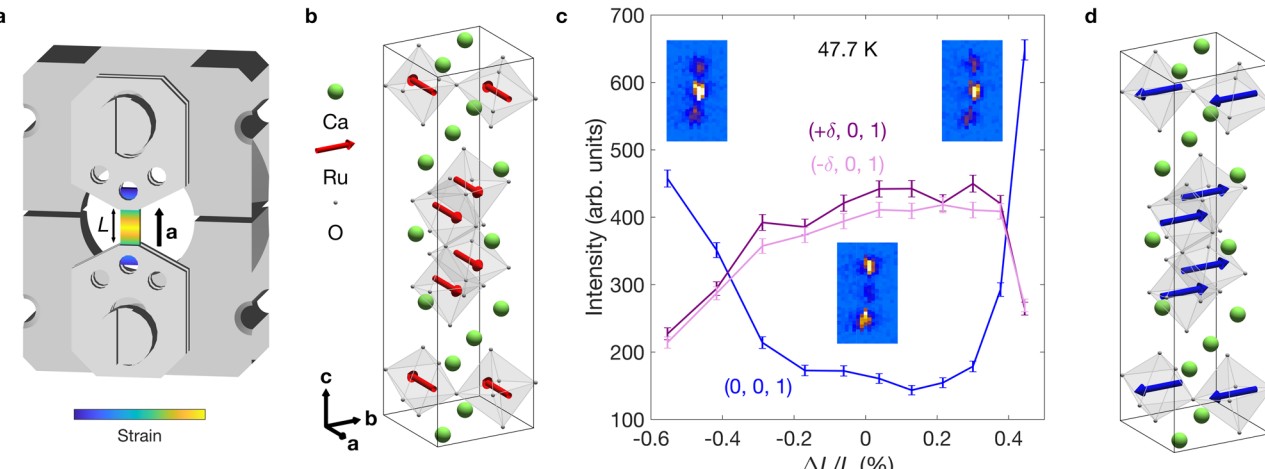

**Fig. 1 | Neutron scattering under stress. a** Schematic of the strain cell used in the neutron scattering experiment. The bridges of the cell and sample plates are shown in grey, and the sample is shown with a blue-to-yellow colourmap to indicate the strain gradient induced along its length. Neutrons are scattered in transmission through the central strained region of the sample. **b** Magnetic structure of $Ca_3Ru_2O_7$ in the $AFM_a$ phase[27]. **c** Integrated intensity of the commensurate $(0, 0, 1)$ and incommensurate $(\delta, 0, 1)$ peaks as a function of applied strain, $\Delta L/L$. The background signal comes from scattering from the strain cell, and errors are standard deviations. The insets show detector images at zero and maximum compressive/tensile applied strain. **d** Magnetic structure of $Ca_3Ru_2O_7$ in the $AFM_b$ phase[27].

be generated by the cell, the maximum strains of $|\Delta L/L| \approx 0.5\%$ are not large enough to fully enter the collinear phases. Finally, and most significantly, the scattering geometry necessitated by the strain cell blocks access to any nuclear Bragg peaks with a finite component along the stress direction. This precludes a determination of the lattice parameters, and therefore the true strain generated in the sample, forcing us to rely on the displacement of the sample plates as a measure of the applied strain. As we will see in the following section, the fact that the epoxy is significantly softer than the sample leads to only a fraction of the applied strain being transmitted to the sample.

## Response of the lattice to applied stress

A full characterisation of the response of the lattice to applied stress is crucial to understanding the magnetic response, as highlighted by previous contradictory reports of the electronic and magnetic changes under uniaxial pressure in Ca₃Ru₂O₇[37,38]. To achieve this, we turned to synchrotron x-ray scattering at beamline I16 of the Diamond Light Source. Size limitations of the closed-cycle cryostat required use of a smaller strain cell than the neutron measurements. The lower stresses are mitigated, however, by the high flux and small focus of the x-ray beam, allowing the use of smaller samples to achieve applied strains over 2%. To maximise beam access we mounted the samples on top of raised sample plates, as shown in Fig. 2a (further details can be found in the Methods). Two samples were measured, one with stress applied along **a** and the other along **b**.

We tracked the positions of multiple structural Bragg peaks while applying compressive stresses. Figure 2b shows representative $2\theta$ scans of the (1, 0, 7) peak for stress applied along **b**. The shifting of the peak arises from the changing lattice parameters, from which we can determine the true strain, $\Delta b/b$. This is shown in Fig. 2c, where we see a linear, temperature-insensitive dependence on $\Delta L/L$, with around 40% of the applied strain transmitted to the sample. We were unable to apply tensile strains due to the asymmetric sample mounting, which also induced a slight bending of the sample under compression (see the Supplementary Information). The bending did not induce significant strain gradients through the probed region, however, as evidenced by the minimal broadening of the peaks in Fig. 2b.

Our x-ray measurements also enable us to quantify the strains along the other crystalline axes, $\varepsilon_x = \Delta x/x$, that arise from the uniaxial stresses applied by the cell, and thereby calculate the Poisson ratios $\nu_{xy} = -\varepsilon_y/\varepsilon_x$. The results are plotted as a function of temperature in Fig. 2d. The anisotropy of the in-plane ($\nu_{ab} \approx \nu_{ba} \approx 0.5$) and out-of-plane ($\nu_{ac} \approx \nu_{bc} \approx 0.2$) Poisson ratios is as expected for a layered material like Ca₃Ru₂O₇.

We can also use x-ray scattering to probe the magnetic phases by exploiting the resonant scattering enhancement on tuning the incident x-ray energy to the Ru $L_2$ edge (2.967 KeV). Alongside the positions of the structural peaks, we monitored the intensities of the ($\delta$, 0, 5) and (0, 0, 5) magnetic peaks at each strain value, the latter at two orthogonal azimuthal angles to differentiate between the AFM$_a$ and AFM$_b$ phases[28]. Repeating these strain sweeps at a range of temperatures enabled us to construct the phase diagrams shown in Fig. 3a, b for stress applied along **a** and **b** respectively (further details of how the phase diagrams were constructed can be found in the Supplementary Information).

The phase diagrams show the transition temperatures varying linearly with compressive strain along both **a** and **b**. The data quality is worse for the sample with stress applied along **a**, most likely due to the poorer crystalline quality of the sample causing a heterogeneous distribution of strain (see the Supplementary Information). Despite this, we can still determine a rate of change of the transition temperatures of ~40 K per percent strain along **a**. The transition temperatures change at a slower rate for the sample stressed along **b**, at around 7 K per percent strain along **b**. Although we could not access the tensile sides of the phase diagrams in our x-ray experiments, our neutron results suggest that the trends should continue unchanged. It is interesting to note that the phase boundaries move in the same direction in both phase diagrams, despite the opposite in-plane deformations of the lattice (as depicted in the insets of Fig. 3a, b).

## A strain-coupled electronic model

To understand the role of strain in the SRT, we developed a phenomenological model in which strain tunes the electronic hopping between nearest-neighbour Ru orbitals. In the spirit of developing a minimal model that captures the physics under study, we included only the Ru $d_{xz}$ and $d_{yz}$ orbitals in a perovskite monolayer. Such a two-band model is motivated by ARPES studies that show that the $d_{xy}$ orbital does not contribute to the Fermi surface[33]. We consider a Hamiltonian $\hat{H} = \hat{H}_t + \hat{H}_U + \hat{H}_\lambda - \mu\hat{N}$. $\hat{H}_t$ is a tight-binding Hamiltonian fitted to ARPES data[33] (see the Methods). $\hat{H}_U = U\sum_{i\tau\alpha}\hat{n}_{i\tau\alpha\uparrow}\hat{n}_{i\tau\alpha\downarrow}$ is the on-site intra-orbital Hubbard interaction, where the lattice site is labelled $i$, the orbital is labelled $\alpha \in \{xz, yz\}$, and $\tau \in \{A, B\}$ is a sublattice index that is introduced to allow for the staggered octahedral tilting. The interaction is treated in the Hartree-Fock mean-field approximation, and all other local electron-electron interactions are neglected for simplicity. The Hamiltonian is then a function of site-averaged charge, $\rho_{\tau\alpha}$, and magnetisation, $\mathbf{M}_{\tau\alpha}$, fields for each orbital and sublattice. $\hat{H}_\lambda$ is the spin-orbit interaction[39],

$$\hat{H}_\lambda = -\frac{i\lambda}{2}\sum_{i,\tau}\left(c^\dagger_{i\tau,xz}\sigma^\tau_z c_{i\tau,yz} + \text{h.c.}\right), \qquad (1)$$

where $c^\dagger_{i\tau\alpha} = (c^\dagger_{i\tau\alpha\uparrow}, c^\dagger_{i\tau\alpha\downarrow})$. This is an on-site term that couples the two orbitals. The octahedral tilting enters the spin-orbit term by rotating the spin quantisation axes on the sublattice sites by $\pm\theta$ about the $b$-axis, as $\sigma^{A(B)}_z = \cos(\theta)\sigma_z \pm \sin(\theta)(\sigma_x - \sigma_y)/\sqrt{2}$. The octahedral rotation (as well as the polar distortion that leads to Rashba-like spin-orbit effects) does not enter the Hamiltonian explicitly, although it does influence the hopping as described in the next section.

Strain enters the model via a phenomenological field, $\varepsilon$, that couples to the nearest-neighbour hopping, $t$, of the tight-binding model (see the Methods for further details). We take a minimal

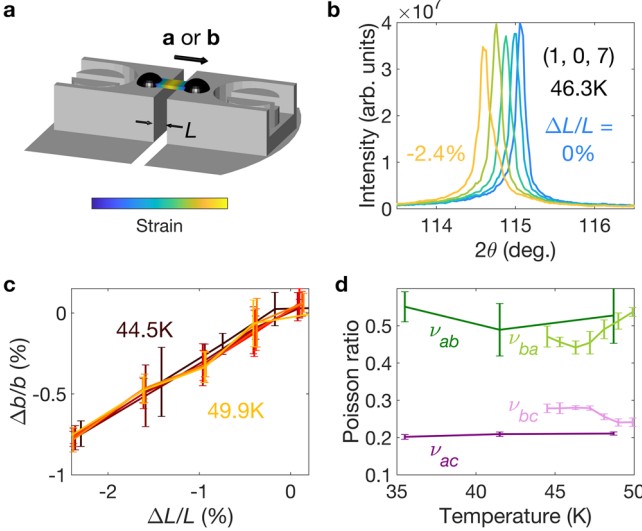

**Fig. 2 | X-ray scattering under stress. a** Schematic of the strain cell used for x-ray scattering. The principles of operation are the same as the neutron setup, except that the sample is mounted on top of raised sample plates to give a large sphere of access for the incident and scattered beams. **b** $2\theta$ scans of the structural (1, 0, 7) Bragg peak at various applied strains. **c** True strain, $\Delta b/b$, as a function of applied strain, $\Delta L/L$, at a range of temperatures through the SRT. **d** Temperature dependences of the Poisson ratios determined from the relative strains along orthogonal axes. All errors are standard deviations.

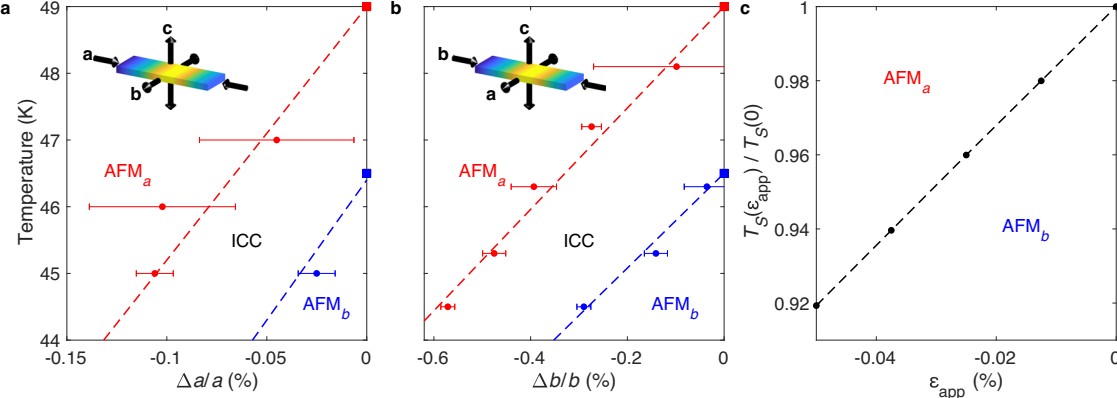

**Fig. 3 | Temperature-strain phase diagrams. a, b** Experimental phase diagrams for stress applied along the *a*- and *b*-axes respectively. The points with errors (standard deviations) are transition temperatures extracted from the strain measurements, and the squares indicate the zero-strain transition temperatures from ref. 28. The dashed lines are guides to the eye. The insets indicate the tensile strains induced along the orthogonal directions according to the Poisson ratios. **c** Theoretical phase diagram under applied strain $\varepsilon_{\mathrm{app}}$, calculated by self-consistently minimising the free energy in Eq. (2) using the parameter values $U = 8$, $\lambda = 0.5$, $\theta = 15°$, $\mu = 8.5$, $\kappa = 400$, $\nu = 12$, $t_0 = 1.12$ and $\varepsilon_0 = 0.066$.

coupling of $t = t_0 + \nu\varepsilon$, where $t_0$ and $\nu$ are parameters, which is valid when the strain magnitude is small. The coupling of strain to further-neighbour hopping is neglected. We also introduce a strain cost set by a parameter, $\kappa$, in the free energy

$$
F(\mathbf{M}, \rho, \varepsilon, \varepsilon_{\mathrm{app}}) = -T \sum_{n,\mathbf{k}} \ln\left(1 + e^{-(\epsilon_{n\mathbf{k}} - \mu)/T}\right) \\
+ U \sum_{\tau,\alpha} \left(\mathbf{M}_{\tau\alpha}^2 - \rho_{\tau\alpha}^2\right) + \frac{1}{2}\kappa\left(\varepsilon_0 + \varepsilon - \varepsilon_{\mathrm{app}}\right)^2, \quad (2)
$$

where $\epsilon_{n\mathbf{k}}$ are the eigenvalues of the electronic Hamiltonian, $\mu$ is the chemical potential, and $\varepsilon_0$ and $\varepsilon_{\mathrm{app}}$ are parameters. The last term is the contribution of the non-electronic degrees of freedom to quadratic order. Within this parameterisation, $\varepsilon$ is the strain measured relative to the unstrained system which we define, via our choice of value of $\varepsilon_0$ and without loss of generality, as the system at the Néel temperature. This is a natural reference point for studying the magnetic phase diagram. As well as the internal strain $\varepsilon$, we include an externally applied strain $\varepsilon_{\mathrm{app}}$ that moves the minimum of the strain cost function away from zero. The electronic state is determined at a particular temperature and externally applied strain by minimising the free energy with respect to the electronic degrees of freedom, $\mathbf{M}_{\tau\alpha}$ and $\rho_{\tau\alpha}$, and the strain field, $\varepsilon$, in a self-consistent manner.

In the absence of strain, $\kappa = \nu = 0$, the solution of this model exhibits easy *a*- and easy *b*-axis ferromagnetism that corresponds to the $\mathrm{AFM}_a$ and $\mathrm{AFM}_b$ phases in the full structure. This results from the interplay of the on-site interaction and spin-orbit coupling, which generate the magnetic anisotropy, and the octahedral tilting that breaks its symmetry in the **a**–**b** plane. The anisotropy is strongly dependent on the nearest-neighbour hopping, and an **a** → **b** reorientation of the ferromagnetic order is achieved by increasing $t$ and therefore the electronic bandwidth. Although our minimal model cannot reproduce the cycloid that mediates the SRT, we would expect this to appear naturally when the full bilayer is considered, due to a uniform Dzyaloshinskii-Moriya interaction that becomes dominant close to the reorientation transition[28].

With a finite $\kappa$ and $\nu$, we find an internal strain being generated self-consistently even with zero applied strain, $\varepsilon_{\mathrm{app}} = 0$, as shown in Fig. 4. We see $\varepsilon$ increase as the temperature is reduced through the $\mathrm{AFM}_a$ phase, lowering the energy of the electronic system at the expense of an elastic energy cost. When the hopping reaches a threshold value the SRT is triggered. This transition is first order and

results in a small positive jump in the strain (see zoomed region in Fig. 4) as well as a jump in the magnetisation. We also capture some of the changes in the Fermi surface that are seen in ARPES studies. We find a continuous Fermi surface reconstruction (insets in Fig. 4) that is not directly driven by the transition, but instead by the increase in magnetisation (and Stoner gap) as temperature is reduced. We emphasise that in this self-consistent calculation, the SRT and Fermi surface reconstruction are achieved as a function of temperature without varying the band filling. The model does not capture the partial gapping of the Fermi surface that is also seen by ARPES[31,33].

To make contact with our strain experiments, we also solve the model with a negative applied strain, $\varepsilon_{\mathrm{app}} < 0$. This reduces the self-consistent strain, $\varepsilon$, and thus the nearest-neighbour hopping, resulting in a reduction in the critical temperature. As shown in Fig. 3c, the linear suppression of the transition temperature matches the experimental phase diagrams for compressive stress applied along **a** and **b** in Fig. 3a, b. As well as reproducing the main sequence of magnetic phases with temperature, our minimal model is therefore able to capture the qualitative features of our strain measurements.

## Microscopic understanding of the transition

Our theoretical model is able to reproduce the phenomenology of the SRT by introducing a strain field that couples to the nearest-neighbour hopping. For a full understanding of the mechanism behind the transition, however, we need to connect this strain field to microscopic distortions of the $\mathrm{Ca_3Ru_2O_7}$ lattice. Given their strong influence on the ground states across the Ruddlesden-Popper series of ruthenates, the octahedral tilt and rotation degrees of freedom (depicted in Fig. 5a, b) are the most obvious candidates.

To be able to identify the tilt and rotation modes with the strain field of the model, we must first show that they directly influence the nearest-neighbour hopping. This can be achieved with a heuristic model of the overlap between neighbouring Ru *d* orbitals, assuming perfectly rigid $\mathrm{RuO_6}$ octahedra. Considering the tilt and rotation modes separately (valid for small distortions), we associate them with angles $\theta$ and $\phi$ respectively. In a similar manner to the two-centre method of Slater and Koster[40], the orbitals on a distorted Ru–O–Ru bond are expanded in a basis of orbitals quantised with respect to an undistorted bond axis (further details can be found in the Methods). We find that the intra-orbital hopping is equal along both nearest-

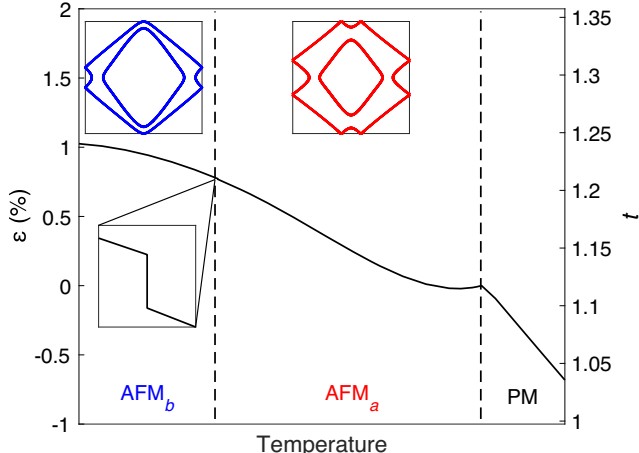

**Fig. 4 | Self-consistent internal strain.** Strain field, $\varepsilon$, as a function of temperature, with the magnetic phase transitions indicated by vertical dashed lines. The corresponding change in the effective nearest-neighbour hopping parameter, $t$, is measured on the right-hand axis. The zoomed region shows a small discontinuity in $\varepsilon$ across the SRT of magnitude $|\Delta\varepsilon| \sim 0.005\%$. The insets show Fermi surfaces calculated in the two magnetic phases. The parameter values are the same as in Fig. 3c. The calculated value of the hopping parameter at the transition corresponds to $U/t \sim 6.6$, which is consistent with values of $t$[33] and $U$[50] previously fitted to the observed electronic structure of $Ca_3Ru_2O_7$.

neighbour bonds, with angle-dependent hopping parameters

$$t(\theta) = \frac{1}{4}\left[\cos(2\sqrt{2}\theta) + 4\cos(\sqrt{2}\theta) - 1\right](dpd\pi)$$
$$+ \cos^3(\sqrt{2}\theta)(dd\pi) + \frac{3}{4}\sin^2(\sqrt{2}\theta)(dd\sigma)$$
$$+ \frac{1}{4}\sin^2(\sqrt{2}\theta)\cos(2\sqrt{2}\theta)(dd\delta)$$
$$t(\phi) = (dpd\pi) + \cos^2(\phi)(dd\pi) + \sin^2(\phi)(dd\delta)$$

(3)

where $(ll'm)$ are Slater-Koster integrals that describe direct hopping between the $d$ orbitals, and $(dpd\pi)$ is the $\pi$-bonding integral that describes indirect hopping via an oxygen $p$ orbital on an undistorted bond. We see that increasing the rotation reduces the contribution from the direct $\pi$-bonding while allowing a weaker $\delta$-bond to develop, therefore reducing the effective hopping. The effect of tilt on the hopping depends on the precise hierarchy of bonding integrals, but we expect from the behaviour of the $(Sr_{1-x}Ca_x)_3Ru_2O_7$ series that increasing the tilt will also reduce the hopping[41–43].

The importance of the octahedral tilts and rotations in the transition is further supported by (zero-stress) measurements of the $Ca_3Ru_2O_7$ structure as a function of temperature. Figure 5c shows the lattice parameters determined by resonant x-ray scattering. The in-plane lattice parameters are seen to increase and $c$ to decrease on cooling through the SRT, all by around 0.05%. The lack of significant changes either side of the SRT points to the low phonon populations at these temperatures, and therefore the importance of the electronic coupling in the structural changes. We also directly determined the symmetry-adapted tilt ($X_3^-$) and rotation ($X_2^+$) mode amplitudes from neutron diffraction measurements, the results of which are shown in Fig. 5d. On cooling through the SRT, the tilts increase by around 0.6% while the rotations decrease by around 1%. These changes are an order of magnitude larger than the changes in bond lengths[30], supporting our earlier assumption that the octahedra remain rigid. Based on our orbital-overlap argument above, we would expect the rotations to increase the hopping on cooling, slightly offset by the increase in tilt. This is in agreement with our model, in which the self-consistent strain and thus the hopping increase with decreasing temperature. We

therefore have a clear picture of how the transition proceeds with temperature: an internal strain, corresponding to changes in the octahedral tilts and rotations, arises through feedback with the electronic system to modulate the hopping and thereby alter the magnetic anisotropy.

Finally, we can apply our understanding of the effects of the tilts and rotations to our strain experiments. While the precise changes in tilt and rotation angle under applied stress are unknown, we can make some general deductions from our knowledge of the Poisson ratios (ignoring strain along **c** which is likely to be mostly accommodated within the CaO rock-salt layers). When compressing along **a**, the dominant effect is likely to be an increase in the tilt angle to accommodate the increased orthorhombicity, together with a smaller reduction in the rotation angle to account for the accompanying expansion along **b**. This leads to a reduction in hopping (offset by the reduced rotation) which suppresses the transition temperature in agreement with Fig. 3a. By contrast, the dominant effect when compressing along **b** is likely to be an increase in the rotation angle. This alone would cause an equal contraction along **a**, such that a significant decrease in the tilt is needed to allow an expansion along **a** according to the Poisson ratio. This could explain the slope of the transition temperature in Fig. 3b if the net effect is still a reduction in the hopping due to the increased rotation, but one that is largely offset by the reduction in tilt.

## Discussion

In this work, we have demonstrated that the spin orientation in $Ca_3Ru_2O_7$ can be controlled with applied stress. By isolating the lattice degrees of freedom, our measurements have provided a unique window into the role of strain in a coupled structural, electronic and magnetic phase transition. Neutron scattering gave us access to the bulk magnetic phases, while resonant x-ray scattering enabled us to probe in detail how strain is transmitted to the lattice, and thereby construct temperature-strain phase diagrams. These experiments motivated a theoretical model in which a strain field couples to the electronic hopping, and drives a spin reorientation when it reaches a critical value.

Our combination of experiment and theory has therefore uncovered a mechanism whereby strain, either externally induced or thermally generated through feedback with the electronic system, drives a global reorientation of the spins and concomitant reconstruction of the Fermi surface in $Ca_3Ru_2O_7$. Given the ubiquity of the octahedral structural unit, we expect this mechanism to be relevant to other transition-metal oxides. For the Ruddlesden-Popper ruthenates in particular, octahedral tilts and rotations have a profound effect on the ground state, whether that is a Mott insulator or unconventional superconductor. Strain then offers the possibility of continuously tuning between these phases without the disorder introduced by chemical doping, and possibly uncovering phases not seen in the ambient structures.

## Methods
### Samples
Single crystals of $Ca_3Ru_2O_7$ were grown using the floating zone method and characterised by wavelength-dispersive x-ray spectrometry, resistivity and magnetization measurements, and x-ray and neutron diffraction[28,29]. The strain axis was aligned with Laue diffraction. A polarised-light microscope was used to identify orthorhombic twin domains, and single-domain pieces were then cut out using a wire saw. The single-domain nature of the samples was checked during the scattering measurements. The samples were found to naturally cleave during cutting, producing long, thin bars with clean (0, 0, 1) faces for scattering (and precluding measurements with stress applied along **c**). The dimensions of the sample used for neutron scattering were $L \times W \times T \approx 1 \times 0.3 \times 0.05$ mm, for x-ray scattering with stress along **a** were

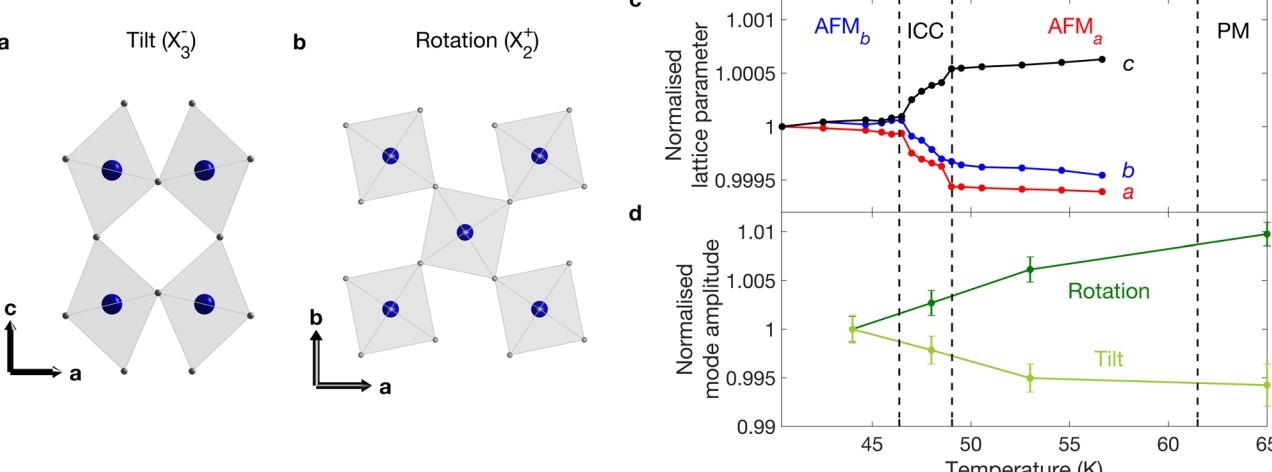

**Fig. 5 | Octahedral tilt and rotation modes. a** Schematic of the tilt mode, consisting of staggered tilts of the $RuO_6$ octahedra around the *b*-axis. **b** Schematic of the rotation mode, consisting of staggered rotations of the octahedra around the *c*-axis. **c** Temperature dependence of the lattice parameters of $Ca_3Ru_2O_7$ from x-ray scattering, normalised to the values at 40.4 K. **d** Temperature dependence of the octahedral rotation and tilt amplitudes determined from a symmetry analysis of single-crystal neutron diffraction data, normalised to the values at 44 K. Errors are standard deviations.

0.3 × 0.1 × 0.04 mm, and for x-ray scattering with stress along **b** were 0.2 × 0.1 × 0.03 mm.

## Neutron scattering under stress

Neutron scattering measurements under stress were performed at the WISH instrument of the ISIS Neutron and Muon Source[36]. We used the CS200T strain cell from Razorbill Instruments[35]. The cell incorporates a 90° access cone allowing transmission measurements with minimal background signal, and a capacitive displacement sensor using which we can calculate the applied strain. The cell was mounted on a custom cryostat stick with feedthroughs for the cables to power the cell and measure the capacitance of the displacement sensor. Voltage was applied to the cell with an RP100 power supply from Razorbill Instruments, and the capacitance was measured with a Keysight E4980AL LCR meter, both of which were integrated into the instrument control software to enable automated control and data acquisition. A temperature sensor was thermally contacted with the body of the cell to give accurate readings of the sample temperature. Cadmium shielding was used to reduce background scattering from the bridges of the cell and from unstrained regions of the sample near the sample plates, while still allowing full beam access in the horizontal plane. The sample was sandwiched between two pairs of sample plates and secured with Stycast 2850FT epoxy, such that **a** lies along the stress direction and vertical detector banks. The sample was cooled without any applied stress, and held at fixed temperatures during the strain sweeps. Data analysis was performed with MANTID[44]. All data are normalised to the cumulative current and a beam monitor. The intensities were obtained by diffraction focussing a small area on the detector around the peak, and then integrating the resulting time-of-flight spectra.

## X-ray scattering under stress

X-ray scattering measurements were performed at beamline I16 of the Diamond Light Source, with the incident energy tuned to the Ru $L_2$ edge (2.967 KeV) by performing 4 bounces on the monochromator and using an extended helium-filled beam pipe. Here, we used the CS100 strain cell from Razorbill Instruments[35], which is smaller than the CS200T and necessitates a reflection scattering geometry. A custom mount was designed to hold the cell in the

closed-cycle cryocooler, with feedthroughs added for the power and capacitance cables. As at WISH, the RP100 power supply and E4980AL LCR meter were interfaced with the beamline software to enable remote control and scripting of the measurements. A Cernox temperature sensor was thermally contacted with the body of the cell. The sample was mounted to the raised sample plates with Stycast 2850FT epoxy, with either **a** or **b** along the stress direction. No top-plates were used in order to maximise beam access, but the asymmetric mounting resulted in stress being transmitted mostly through the lower surface of the sample causing it to bend (see the Supplementary Information). This, along with the difference in elastic moduli of the sample (~100 GPa) and epoxy (~4 GPa), leads to the true strain being significantly lower than the applied strain. The diffractometer was operated with a vertical scattering geometry in fixed-azimuth mode, and the scattered intensity was measured with an in-vacuum Pilatus 100K area detector in ultrahigh gain mode. All temperature changes were conducted with zero applied stress, and the temperature was kept constant during strain measurements. Data analysis was carried out using the PY16 program[45], and reciprocal-space maps were reconstructed from rocking scans using the MILLERSPACEMAPPER software[46]. Intensities were obtained by summing over a region-of-interest on the area detector and fitting the resulting rocking curves with pseudo-Voigt profiles plus a constant background.

## Neutron diffraction

The neutron diffraction data shown in Fig. 5d were taken on the four-circle diffractometer D9 at the Institut Laue Langevin. A wavelength of 0.836 Å was obtained from a Cu(220) monochromator giving access to 800 reflections (up to $h = 10, k = 10, l = 15$). The symmetry mode analysis was performed using the AMPLIMODES software[47] from the Bilbao Crystallographic Server, considering the high-symmetry parent phase $I4/mmm$ (139) and the low symmetry phase $Bb2_1m$ (36). The mode amplitudes were directly refined at each temperature by the least-squares method using FULLPROF[48].

## Strain-coupled electronic model

We apply a tight-binding model for $Ca_3Ru_2O_7$ that has been fitted to ARPES data[33]. The tight-binding Hamiltonian in a crystal momentum

basis is

$$
\begin{aligned}
\hat{H}_t(\mathbf{k}) = \sum_\tau \big[ & \epsilon_x(\mathbf{k})\, \underline{c}^\dagger_{xz,\tau}(\mathbf{k})\, \sigma_0\, \underline{c}_{xz,\bar\tau}(\mathbf{k}) \\
& + \epsilon_y(\mathbf{k})\, \underline{c}^\dagger_{yz,\tau}(\mathbf{k})\, \sigma_0\, \underline{c}_{yz,\bar\tau}(\mathbf{k}) \big] \\
& + \sum_{\alpha,\tau} \big[ \epsilon_{ab}(\mathbf{k})\, \underline{c}^\dagger_{\alpha,\tau}(\mathbf{k})\, \sigma_0\, \underline{c}_{\alpha,\tau}(\mathbf{k}) \\
& + \epsilon'_{ab}(\mathbf{k})\, \underline{c}^\dagger_{\alpha,\tau}(\mathbf{k})\, \sigma_0\, \underline{c}_{\bar\alpha,\tau}(\mathbf{k}) \big]
\end{aligned}
\tag{4}
$$

where $\underline{c}^\dagger_{\alpha\tau}(\mathbf{k}) = (c^\dagger_{\alpha\tau\uparrow}(\mathbf{k}), c^\dagger_{\alpha\tau\downarrow}(\mathbf{k}))$, $\sigma_0$ is the $2\times2$ identity matrix in spin space, $\alpha \in \{xz, yz\}$ and $\tau \in \{A, B\}$. Bars are used to denote flipped orbital and sublattice indices, i.e. $\overline{xz} = yz$, $\bar{A} = B$, and vice versa. In the tetragonal crystal momentum basis, $\mathbf{k} = (k_x, k_y)$, the dispersions are

$$
\begin{aligned}
\epsilon_x(\mathbf{k}) &= -2t\cos(k_x a) \\
\epsilon_y(\mathbf{k}) &= -2t\cos(k_y a) \\
\epsilon_{ab}(\mathbf{k}) &= -2t_a\cos((k_x - k_y)a) - 2t_b\cos((k_x + k_y)a) \\
\epsilon'_{ab}(\mathbf{k}) &= -2t'_a\cos((k_x - k_y)a) - 2t'_b\cos((k_x + k_y)a)
\end{aligned}
\tag{5}
$$

where $k_{x,y} \in [-\frac{\pi}{a}, \frac{\pi}{a}]$ with lattice constant $a$. $\epsilon_{x(y)}$ describes the nearest-neighbour hopping between $xz(yz)$ orbitals along the $x(y)$ direction, and $\epsilon_{ab}$ and $\epsilon'_{ab}$ describe the intra- and inter-orbital next-nearest-neighbour hopping along the $a$ and $b$ orthorhombic directions. We use $t_a = 0.04\tilde{t}$, $t_b = 0.16\tilde{t}$ and $t'_a = -t'_b = 0.079\tilde{t}$, with $\tilde{t} = 1.20$. The coefficients were chosen by fitting the Fermi surfaces in both magnetic phases to those measured using ARPES[33]. Only the intra-orbital nearest-neighbour hopping parameter couples to the strain field, as $t = t_0 + \nu\varepsilon$. Strictly, the octahedral tilting and rotation also induce inter-orbital nearest-neighbour hopping between the $d_{xz}$ and $d_{yz}$ orbitals[39]. This hopping mode is neglected in ref. 33, and we also neglect it as a sub-leading effect of the lattice distortions.

In our minimal model, we consider only the on-site intra-orbital repulsion, $\hat{H}_U = U\sum_{i\tau\alpha} \hat{n}_{i\tau\alpha\uparrow} \hat{n}_{i\tau\alpha\downarrow}$, and neglect the inter-orbital repulsion and Hund's coupling for simplicity. $\hat{H}_U$ is treated in the Hartree-Fock mean-field approximation by decoupling in the charge, $\hat{\rho}_{i\tau\alpha} = \frac{1}{2}\underline{c}^\dagger_{i\tau\alpha}\sigma_0\underline{c}_{i\tau\alpha}$, and spin, $\hat{\mathbf{S}}_{i\tau\alpha} = \frac{1}{2}\underline{c}^\dagger_{i\tau\alpha}\boldsymbol{\sigma}\underline{c}_{i\tau\alpha}$, channels, where $\boldsymbol{\sigma} = (\sigma_x, \sigma_y, \sigma_z)$ is a vector of Pauli matrices. Defining $\rho_{\tau\alpha}$ and $\mathbf{M}_{\tau\alpha}$ as the site-averaged charge and spin densities, the interaction decouples to

$$
\begin{aligned}
\hat{H}^{\mathrm{MF}}_U = & U\sum_{i,\tau,\alpha} \underline{c}^\dagger_{i\tau\alpha} (\rho_{\tau\alpha}\sigma_0 - \mathbf{M}_{\tau\alpha}\cdot\boldsymbol{\sigma})\underline{c}_{i\tau\alpha} \\
& + U\sum_{\tau,\alpha} \left( \mathbf{M}^2_{\tau\alpha} - \rho^2_{\tau\alpha} \right).
\end{aligned}
\tag{6}
$$

The system is described in the mean-field and at fixed chemical potential, $\mu$, by the Hamiltonian, $\hat{H}^{\mathrm{MF}} = \hat{H}_t + \hat{H}_\lambda + \hat{H}^{\mathrm{MF}}_U - \mu\hat{N} = \sum_{n\mathbf{k}}(\epsilon_{n\mathbf{k}} - \mu)\hat{n}_{n\mathbf{k}} + U\sum_{\tau\alpha}(\mathbf{M}^2_{\tau\alpha} - \rho^2_{\tau\alpha})$. The free energy is evaluated via the partition function to give Eq. (2) (in the absence of coupling to strain, $\nu = \kappa = 0$) in terms of the energy eigenvalues, $\epsilon_{n\mathbf{k}}$, in a band and crystal momentum basis. The strain field is then introduced into the model as described in the Results.

The values of strain parameters $(\nu, \kappa)$ are chosen such that the calculated strain is of the same order of magnitude as seen in experiment (see Figs. 4 and 5c), and the magnitude of strain-hopping coupling, $\nu$, is comparable to a value calculated from first principles for a related material[49]. It is worth noting that the solution of this model is invariant under the scaling of the strain parameters as $\nu \to x\nu$, $\kappa \to x^2\kappa$, $\varepsilon_0 \to \varepsilon_0/x$ and $\varepsilon_{\mathrm{app}} \to \varepsilon_{\mathrm{app}}/x$. This scales the self-consistent strain solution as $\varepsilon \to \varepsilon/x$ but leaves the electronic solution unchanged.

## Effective hopping on a distorted lattice

We calculate the effective nearest-neighbour hopping between ruthenium $d$ orbitals in the presence of octahedral tilting and

rotational distortions (separately), considering the direct orbital overlap and the indirect overlap via oxygen $p$ orbitals.

The hopping is first defined in the absence of lattice distortions, i.e. for a straight Ru–O–Ru bond. In the case of hopping due to direct $d$ orbital overlap, we apply the Slater-Koster two-centre approximation,

$$
t^{\mathrm{Ru-Ru}}_{lm,l'm'} = \left\langle \psi^{\mathrm{Ru}}_{lm}(\mathbf{r}) \middle| \hat{H}^{\mathrm{Ru-Ru}} \middle| \psi^{\mathrm{Ru}}_{l'm'}(\mathbf{r} + R\hat{\mathbf{x}}) \right\rangle
\tag{7}
$$

where $\psi^{\mathrm{Ru}}_{lm}$ is an atomic orbital centred on a ruthenium ion whose angular momentum quantisation axis is dictated by the crystal field, and $\hat{H}^{\mathrm{Ru-Ru}}$ is the sum of the kinetic energy operator and ionic potentials on the bond. In the case of indirect hopping, the effective hopping is calculated as a second order process in which an electron or hole hops between ruthenium $d$ orbitals via an oxygen $p$ orbital. The hopping matrix element for a straight Ru–O–Ru bond is

$$
t^{\mathrm{Ru-O-Ru}}_{lm,l'm'} = -\sum_{l',m'} \frac{\left\langle \psi^{\mathrm{Ru}}_{lm}(\mathbf{r}) \middle| \hat{H}^{\mathrm{Ru-O}} \middle| \psi^{\mathrm{O}}_{l'm'}(\mathbf{r} + \frac{R}{2}\hat{\mathbf{x}}) \right\rangle \left\langle \psi^{\mathrm{O}}_{l'm'}(\mathbf{r} + \frac{R}{2}\hat{\mathbf{x}}) \middle| \hat{H}^{\mathrm{O-Ru}} \middle| \psi^{\mathrm{Ru}}_{lm}(\mathbf{r} + R\hat{\mathbf{x}}) \right\rangle}{E_{lm} - E'_{l'm'}}
\tag{8}
$$

where $E_{lm}$ and $E'_{l'm'}$ are the total energies of the system when the electron/hole occupies a ruthenium orbital, $\psi^{\mathrm{Ru}}_{lm}$, or an oxygen orbital, $\psi^{\mathrm{O}}_{l'm'}$, respectively. The numerator is a product of matrix elements for Ru–O and O–Ru hopping processes. We make a key assumption that the lattice distortions are a weak perturbation on the oxygen $p$ orbitals, which remain degenerate. Every indirect hopping process then has the same intermediate energy, $E'$, and the effect of lattice distortions is only to modify the two-centre integrals in the numerator.

The lattice distortions are introduced by rotating all orbitals in space accordingly. Since the orbital orientation is dictated most strongly by the local (octahedral) crystal field, we assume that the ruthenium $d$ orbitals are rotated/tilted along with the octahedra. Note that we decompose the (small) tilt about $\mathbf{b}$ into tilts of $\theta/\sqrt{2}$ about the $x$- and $y$-axes. We also assume that the change in Ru–Ru distance due to the distortions has a sub-leading effect on the hopping and as such is neglected.

The rotated orbitals are then expanded in a basis of atomic orbitals that are quantised with respect an appropriate bond axis. In the direct case this is the Ru–Ru bond. The effective hopping can then be written in terms of matrix elements for the hopping of an electron between the basis orbitals, which are the Slater-Koster integrals

$$
(ll'm)\delta_{mm'} = \left\langle \phi^{\mathrm{Ru}}_{lm}(\mathbf{r}) \middle| \hat{H}^{\mathrm{Ru-Ru}} \middle| \phi^{\mathrm{Ru}}_{l'm'}(\mathbf{r} + R\hat{\mathbf{x}}) \right\rangle
\tag{9}
$$

where $\phi_{lm}$ is an orbital that is quantised with respect to the bond axis. As is convention, we use the labels $l, l' \in \{s, p, d\}$ and $m \in \{\sigma, \pi, \delta\}$ on the left-hand side. In the indirect case we expand in bases of orbitals that are quantised with respect to the Ru–O and O–Ru bonds. We find that the effective hopping can be written as a function of the hopping along a straight bond,

$$
(ll'm)\delta_{mm'} = -\frac{\left\langle \phi^{\mathrm{Ru}}_{lm}(\mathbf{r}) \middle| \hat{H}^{\mathrm{Ru-O}} \middle| \phi^{\mathrm{O}}_{l'm'}(\mathbf{r} + R\hat{\mathbf{x}}) \right\rangle \left\langle \phi^{\mathrm{O}}_{l'm'}(\mathbf{r} + R\hat{\mathbf{x}}) \middle| \hat{H}^{\mathrm{O-Ru}} \middle| \phi^{\mathrm{Ru}}_{lm}(\mathbf{r} + 2R\hat{\mathbf{x}}) \right\rangle}{E_{lm} - E'_{l'm'}}.
\tag{10}
$$

On the LHS we have introduced a similar notation for the indirect hopping matrix element to that of the Slater-Koster integrals. The numerator is a product of two Slater-Koster integrals.

By summing the direct and indirect hoppings in each case, we find the effective nearest-neighbour hoppings of Eq. (3) (represented graphically in Fig. 6). We find that increasing the rotation angle reduces the contribution from the direct $\pi$-bonding while allowing a weaker $\delta$-bond to develop, therefore reducing the effective hopping. Increasing the tilt angle reduces the contribution from both $\pi$-bonding modes, while allowing $\sigma$- and $\delta$-bonding modes to develop. Since the indirect

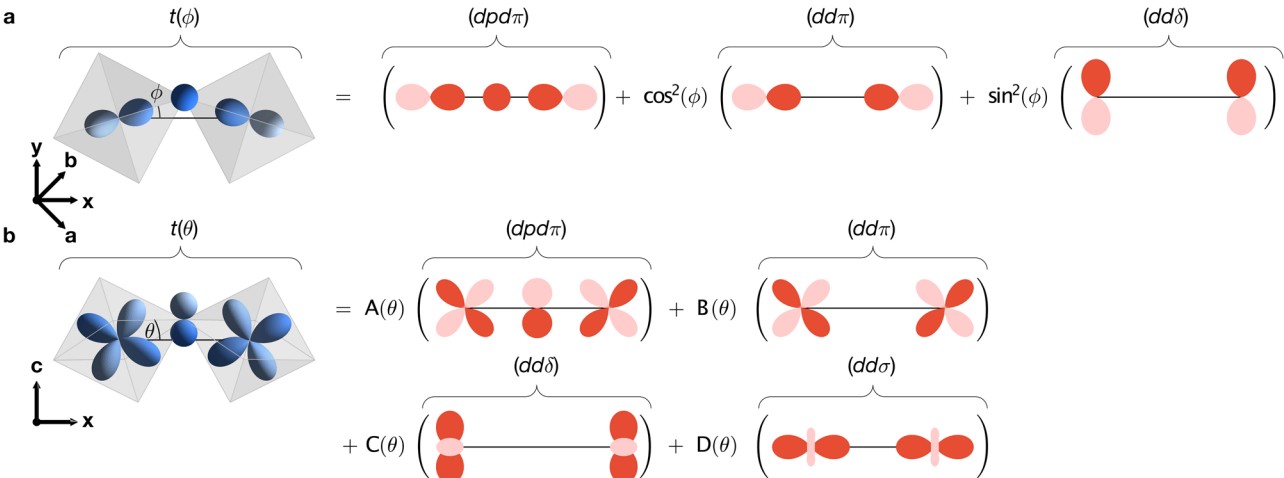

**Fig. 6 | Rotation- and tilt-dependence of hopping. a, b** Dependence of the in-plane nearest-neighbour hopping, $t$, between Ru $d_{xz}$ orbitals along **x** on the octahedral rotation angle, $\phi$, and octahedral tilt angle, $\theta$, respectively. The orbitals on the distorted Ru–O–Ru bond are shown in blue on the left side of each equation. Direct hopping is expressed in terms of Slater-Koster integrals, $(ll'm)$, illustrated in red on the right side of each equation. The indirect hopping, due to $\pi$-bonding with oxygen $p$ orbitals, is expressed in terms of $(dpd\pi)$, the $d$–$p$–$d$ $\pi$-bonding integral on a straight Ru–O–Ru bond (also shown in red). All oxygen $p$ orbitals are considered but only the $p_z$ orbital is shown for clarity. The coefficients in (**b**) are $A(\theta) = \frac{1}{4}\left[\cos(2\sqrt{2}\theta) + 4\cos(\sqrt{2}\theta) - 1\right], B(\theta) = \cos^3(\sqrt{2}\theta), C(\theta) = \frac{1}{4}\sin^2(\sqrt{2}\theta)\cos(2\sqrt{2}\theta)$ and $D(\theta) = \frac{3}{4}\sin^2(\sqrt{2}\theta)$. Identical results are obtained for the hopping between $d_{yz}$ orbitals along **y**.

$\pi$-bonding, $(dpd\pi)$, likely dominates, we expect tilting to cause a net reduction in the effective hopping.

## Data availability
Raw data from the neutron scattering measurements under stress are available at https://doi.org/10.5286/ISIS.E.RB1920210. Raw data from the neutron diffraction measurements are available at https://doi.org/10.5291/ILL-DATA.EASY-951. Source data used to plot the figures are available at https://doi.org/10.6084/m9.figshare.23508354.

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

## Acknowledgements

We thank Richard Thorogate for assistance with the resistivity measurements, Daniel Nye and Gavin Stenning for assistance with the powder x-ray diffraction and Laue alignment in the Materials Characterisation Laboratory at the ISIS Neutron and Muon Source, Mike Matthews for technical support at I16, and Jacob Simms, Katherine Mordecai, Jon Bones and David Keymer for technical support at WISH. C.D.D. was supported by the Engineering and Physical Sciences Research Council (EPSRC) Centre for Doctoral Training in the Advanced Characterisation of Materials under Grant No. EP/L015277/1. A.H.W. was supported by the EPSRC under Grant No. EP/N509577/1. D.D.K. was supported by the EPSRC under Grant No. EP/W00562X/1. Work at UCL was supported by the EPSRC under Grants No. EP/W005786/1, EP/N027671/1, EP/P013449/1 and EP/N509577/1. Experiments at the ISIS Neutron and Muon Source were supported by beamtime allocation RB1920210 from the Science and Technology Facilities Council. We acknowledge the Diamond Light Source for time on beamline I16 under proposals MM23580 and MM25554. We thank Institut Laue Langevin for access to the neutron diffractometer D9 under proposal EASY-951.

## Author contributions

R.S.P. and C.D.D. grew and prepared the single crystal samples respectively. C.D.D., L.S.I.V., Q.F., R.S.P., R.D.J., P.M., D.D.K. and F.O. performed the neutron scattering measurements under stress, and C.D.D., P.M., D.D.K. and F.O. analysed the data. C.D.D., L.S.I.V., Q.F., R.S.P., R.D.J. and D.G.P. performed the x-ray scattering measurements under stress, and C.D.D. and D.G.P. analysed the data. Q.F., C.V.C. and O.F. performed the neutron diffraction measurements and all analysed the data. A.H.W., M.P.K., F.K. and A.G.G. developed the theoretical model. C.D.D. and A.H.W. wrote the paper, with input from all authors. A.G.G. and D.F.M. oversaw the study.

## Competing interests

The authors declare no competing interests.
