## [Peer Review File · Nature Communications]

Reviewers' Comments:

Reviewer #1:

Remarks to the Author:

The manuscript of Dashwood et al. presents a uniaxial pressure study of Ca₃Ru₂O₇. Both neutron and resonant x-ray scattering results are put forward and thoroughly discussed via theoretical modelling. All-together, this study presents an understanding of how structural lattice tilt and rotational distortions couple to the magnetic spin rotation transition. As such, the manuscript is interesting and timely. However, for publication in Nature Communication several improvements are required.

(1) Unstrained Ca₃Ru₂O₇ crystals are typically twinned. Application of strain would/could detwin the crystal. Since only compressive strain is applied, can the authors prove that compressive a- and b-axis strain are not equivalent? Would the strain axis not always be the short lattice parameter axis?

(2) Connected to (1), it would be useful to indicate at what temperature the strain is applied. Presumably, it is applied at around 40 K where the experiments are carried out?! However, it could also be applied at room temperature in which case a detwinning is more likely. I was not able to find this information in the manuscript. Usually, a Nature Communication article would have a method section where such information is provided. In this manuscript, technical information is given partly in the main text and partly in an appendix. Streamlining would improve the manuscript.

(3) Figure panels 3a,3b are probably the most interesting experimental result. Panel a display compressive a-axis strain and panel b the corresponding b-axis strain. In panel a, the AFM data consist of a single point plus the last end of an error bar. The bit of an error bar might be linked to a data point in panel b where there is an error bar going beyond the panel. Are the authors thinking that compressive b-axis pressure correspond to tensile a-axis pressure?

(4) It is explained why the strain is not extracted from the neutron scattering experiment directly. Even these technical challenges were eliminated, would the momentum resolution of the neutron scattering technique be sufficient to extract say 0.5% strain?

(5) There are a few references that the authors may consider to cite:

(a) Stress induced topological phase transitions – Nat. Mat. 20, 1093 (2021)

(b) Recent neutron and resonant x-ray scattering experiments applying uniaxial strain – arXiv:2204.02304 and Nat. Comm. 13, 1795 (2022).

(c) Transport experiments on detwinned Ca₃Ru₂O₇ is in addition to the cited ARPES work providing evidence of C4 broken Fermi surface structure – PRB 97, 041113(R) (2018).

Reviewer #2:

Remarks to the Author:

Combining the uniaxial strain technique with neutron diffraction (CS200T) and resonant x-ray diffraction (CS100), the authors demonstrated that the spin-reorientation transition in Ca₃Ru₂O₇ can be tuned by uniaxial stress applied on the a/b axis. They use neutron diffraction to track the response of the magnetic order to uniaxial strain and find that, at the spin-reorientation temperature (T=47.7K), compressive (tensile) strain along the a (b) axis drives the system into AFMa (AFMb) order. Moreover, they use resonant x-ray diffraction to study the magnetic and structural response of the sample to applied uniaxial stress simultaneously, determine the actual strain in the sample, and successfully map out the temperature-strain magnetic phase diagram for the uniaxial stress along the a and b axes.

The experimental design is elegant, and the experimental results are of high quality. The authors have also developed a theoretical model to understand the microscopic mechanism of the uniaxial-strain-driven spin reorientation in Ca₃Ru₂O₇, which sounds reasonable.

The uniaxial strain has been suggested to be a powerful controlling parameter in tuning the electronic properties (including intertwined orders) of the quantum materials hosting strong

electron-lattice coupling. The elegant experimental strategy, and comprehensive study presented in this manuscript, can advance this research field. The system, layered Ruddlesden-Popper ruthenate $\text{Ca}_{n+1}\text{RuO}_{3n+1}$, has been a research focus in the past years. Tuning the magnetism via lattice strain in $\text{Ca}_3\text{Ru}_2\text{O}_7$ and understanding the mechanism is very important.

As this study shows that people can tune the intertwined orders in similar systems by affecting the TMO6 (TM, transition metal) octahedra's tilting and rotation. I expect this study will inspire more following works in similar materials.

Given the above-mentioned importance, I believe this study is worthy of publication in Nature Communications. As the manuscript is well-written and organized, I recommend its publication without major changes. I suggest the authors move the Appendix to the "Methods" section or the supplemental materials.

Some other minor comments:

(1) The layered Ruddlesden-Popper ruthenates usually have very large elastic moduli (elastic modulus along the a/b axis could be $>100\text{GPa}$). I guess this could be the reason that the strain transferring efficiency is low to 20% in Fig. S2(c), as the shear modulus for Stycast 2850FT is $\sim 4\text{GPa}$ and the method (burying the sample ends in a droplet of the epoxy without Ti plates) would relax the strain quite a lot. The authors should make this clear in the experimental details.

(2) Such a study is very challenging and relies on an in-depth collaboration with neutron and synchrotron light sources. I suggest that the authors (especially the authors from ISIS and Diamond light source) could establish the uniaxial strain tuning as a standard sample environment for some experienced users.

Reviewer #3:

Remarks to the Author:

The article by Dashwood et al combines a large body of neutron and synchrotron x-ray diffraction work with a theoretical modelling. Both neutron and x-ray experiments were performed under uniaxial strain using 2 different setups from the Razorbill Instruments. The authors chose to study a bilayer ruthenate $\text{Ca}_3\text{Ru}_2\text{O}_7$ with an antiferromagnetic ground state AFM_b, characterized by a propagation vector (001) and an ordered moment along the b-axis of an orthorhombic structure with Bb21m space group. At temperatures above $\sim 49\text{K}$, but below $\sim 60\text{K}$, the moment is oriented along the a-axis. It is the effect of compressive and tensile strain on this spin reorientation transition that was studied in this work. Recently, a Rashba-based hybridization mechanism was proposed to explain the nature of this transition, Ref[27] in the manuscript. The interpretation put forward by Dashwood et al is seemingly at odds with that work. Dashwood et al claim instead that the transition is driven by a lattice strain. To support their claim, they show in Fig.1 results of neutron diffraction at WISH diffractometer. The integrated intensity of magnetic Bragg peaks shows a similar response to both tensile and compressive strain- namely the intensity of the commensurate peak increases with strain whereas the intensity of incommensurate peak decreases. This constitutes the main result of neutron diffraction part. The measurements are reported at $47,7\text{K}$, which corresponds to a region of existence of a magnetic cycloid, in which magnetic moments rotate in the ab-plane. Fig.1 b&d do not correctly represent experimental findings- they show AFM_a & AFM_b commensurate antiferromagnetism regions. The outstanding question is what are the changes to the magnetic structure under strain? The authors promised "Unprecedented insight", but a reader is left guessing about magnetic structure under strain. Further, Fig.1a shows that a significant part of the sample is not strained, but surely contributes to a measured intensity. How does one resolve response from strained and unstrained parts of a crystal? Finally, as authors admit, the cell geometry precludes an access to nuclear Bragg peaks. I salute their honesty, but this indicates that the true strain was not measured in the experiment and the x-axis units in Fig.1c is only a suggestion.

The authors then proceed with a synchrotron X-ray diffraction work under strain. The setup for this experiment is markedly different from the one used in the neutron diffraction. Here, the mounting of the sample is asymmetric and only the bottom part of the crystal is glued to moving parts of the

cell. This experiment was performed under a compressive strain only, but allowed the authors to quantify the true strain experienced by the lattice, which is an important result. They have also tracked intensities of magnetic Bragg peaks and constructed the temperature-strain magnetic phase diagrams for strain applied along the a- and b-directions. Rather large error bars in Fig.3a make a direct comparison to the neutron diffraction work difficult. Applied in both directions the strain lowers the transition temperatures to both AFM_a and AFM_b states. It appears that the region of existence of a cycloid (temperature wise) is not affected by the strain. Doesn't it suggest that the modulation is not coupled the strain? The theoretical model in Fig.3c qualitatively reproduces the experimental results, but contains no ICC region. Crystals of Ca₃Ru₂O₇ cleave easily in the ab-plane. Doesn't this create huge strain inhomogeneity with this type of sample mounting, where the bottom part of the sample is under much higher strain than the top one? The authors should comment on this. Ref.21 reports an observation of ~0,1 mm crystalline domains, in which the a- and b-axes are interchanged. The authors should comment how (if) they ensured that their crystals are in mono-domain state. Otherwise the discussion about anisotropic response of a lattice to a strain is redundant.

Some general comments:

A correct identification of the thermodynamic potential has not been not achieved. It would require to work out a clear link between strain-tensor-components, internal degrees-of-freedom, i.e. the tilts and rotations of RuO₆-octahedra and the electronic system – which is in reality also responsible for the bonding and the moduli of mechanical degrees-of-freedom. Most of the relevant couplings may be non-linear, as the various modes involved (magnetic, electronic/orbital occupation, tilts and rotations vs. strains, i.e. acoustic phonons) have different symmetries. Hence, one has to understand higher-order coupling terms in the thermodynamic potential. Here, all this is executed only implicitly and at a crude level of a tight-binding-model and, hence, the relevance of such coupling as eq.(C2) appears unclear.

Worse, all parameters mentioned seem not to have any units and appear to be meaningless. If the energy units are eV, then for a hopping parameter of $t_0=0.33$ eV an $U=8$ eV seems to be much too large for a reasonable model of the electronic structure in an (almost metallic) ruthenate, although with a stable local moment.

The strains are measured in 'arb. units', but strains are always dimensionless, and should have definite meaning as relative displacement/change of length. In the Fig.4 strain for the self-consistent calculation changes by order-of 0.1-0.2 which makes no sense to me. The experiment show strains like 0.005.

If the model has any meaning, the authors should demonstrate that it can reproduce at least a correct order-of-magnitude for basic mechanical properties (bulk modulus, thermal expansion, elastic constants).

Therefore, based on such a proposal and a schematic model, the authors cannot claim to have identified the correct mechanism. For that, they should make a much more quantitative effort specific to the material. Hence, this is a proposal to rationalize their experimental data, but no resolution of the issue.

Reply to Referees for “Strain control of a bandwidth-driven spin reorientation in $\text{Ca}_3\text{Ru}_2\text{O}_7$ ”

I. FIRST REFEREE

The manuscript of Dashwood et al. presents a uniaxial pressure study of $\text{Ca}_3\text{Ru}_2\text{O}_7$. Both neutron and resonant x-ray scattering results are put forward and thoroughly discussed via theoretical modelling. All-together, this study presents an understanding of how structural lattice tilt and rotational distortions couple to the magnetic spin rotation transition. As such, the manuscript is interesting and timely.

We thank the referee for their review of our manuscript, and are pleased that they recognise the significance of our work.

However, for publication in Nature Communication several improvements are required.

(1) Unstrained $\text{Ca}_3\text{Ru}_2\text{O}_7$ crystals are typically twinned. Application of strain would/could detwin the crystal. Since only compressive strain is applied, can the authors prove that compressive a- and b-axis strain are not equivalent? Would the strain axis not always be the short lattice parameter axis?

The referee is correct to note that as-grown $\text{Ca}_3\text{Ru}_2\text{O}_7$ crystals are generally twinned, which would complicate the analysis of our strain data. To avoid this, we identified twin domains using a polarised light microscope and cut our strain samples from single-domain regions (see Supplemental Material of Ref. [1] for further details). We confirmed the absence of twinning in our samples in both the neutron and x-ray scattering experiments. The maximum strains reached in the experiments are an order of magnitude below the orthorhombicity of the lattice, such that we can always determine the a and b axes unambiguously. We noted the single-domain nature of the strain samples in the appendix of our original submission, but have expanded this in the revised Methods section to make it clearer.

(2) Connected to (1), it would be useful to indicate at what temperature the strain is applied. Presumably, it is applied at around 40 K where the experiments are carried out?! However, it could also be applied at room temperature in which case a detwinning is more likely. I was not able to find this information in the manuscript. Usually, a Nature Communication article would have a method section where such information is provided. In this manuscript, technical information is given partly in the main text and partly in an appendix. Streamlining would improve the manuscript.

We thank the referee for highlighting this point, which was an omission in our original submission. The strain was always applied at the measured temperature and then taken back to zero before any temperature changes. We have reformatted the manuscript to include a full Methods section that contains details of these measurement procedures.

(3) Figure panels 3a, 3b are probably the most interesting experimental result. Panel a display compressive a-axis strain and panel b the corresponding b-axis strain. In panel a, the AFM data consist of a single point plus the last end of an error bar. The bit of an error bar might be linked to a data point in panel b where there is an error bar going beyond the panel. Are the authors thinking that compressive b-axis pressure correspond to tensile a-axis pressure?

We agree with the referee that the previous presentation of the data in Fig. 3a–b was confusing. We have revised the figure to include the zero-strain transition temperatures from our previous study [1], and removed the spurious error bars which were artifacts from the plotting procedure.

(4) It is explained why the strain is not extracted from the neutron scattering experiment directly. Even these technical challenges were eliminated, would the momentum resolution of the neutron scattering technique be sufficient to extract say 0.5% strain?

WISH has a very high angular resolution that is well suited to measuring small shifts in the position of peaks. In a previous experiment (without the strain cell), we were able to determine the a lattice parameter of $\text{Ca}_3\text{Ru}_2\text{O}_7$ with a relative error of around 0.01% [1]. Had the scattering geometry allowed, we are therefore confident that the resolution of WISH would have been sufficient to determine the true strain.

(5) There are a few references that the authors may consider to cite: (a) Stress induced topological phase transitions – Nat. Mat. 20, 1093 (2021) (b) Recent neutron and resonant x-ray scattering experiments applying uniaxial strain

– *arXiv:2204.02304* and *Nat. Comm.* 13, 1795 (2022) (c) *Transport experiments on detwinned $\text{Ca}_3\text{Ru}_2\text{O}_7$ is in addition to the cited ARPES work providing evidence of C_4 broken Fermi surface structure – PRB 97, 041113(R) (2018).*

We thank the referee for drawing our attention to these references, which we have cited in the revised manuscript.

II. SECOND REFEREE

Combining the uniaxial strain technique with neutron diffraction (CS200T) and resonant x-ray diffraction (CS100), the authors demonstrated that the spin-reorientation transition in $\text{Ca}_3\text{Ru}_2\text{O}_7$ can be tuned by uniaxial stress applied on the a/b axis. They use neutron diffraction to track the response of the magnetic order to uniaxial strain and find that, at the spin-reorientation temperature ($T = 47.7\text{ K}$), compressive (tensile) strain along the a (b) axis drives the system into AFM_a (AFM_b) order. Moreover, they use resonant x-ray diffraction to study the magnetic and structural response of the sample to applied uniaxial stress simultaneously, determine the actual strain in the sample, and successfully map out the temperature-strain magnetic phase diagram for the uniaxial stress along the a and b axes.

The experimental design is elegant, and the experimental results are of high quality. The authors have also developed a theoretical model to understand the microscopic mechanism of the uniaxial-strain-driven spin reorientation in $\text{Ca}_3\text{Ru}_2\text{O}_7$, which sounds reasonable.

The uniaxial strain has been suggested to be a powerful controlling parameter in tuning the electronic properties (including intertwined orders) of the quantum materials hosting strong electron-lattice coupling. The elegant experimental strategy, and comprehensive study presented in this manuscript, can advance this research field. The system, layered Ruddlesden-Popper ruthenate $\text{Ca}_{n+1}\text{Ru}_n\text{O}_{3n+1}$, has been a research focus in the past years. Tuning the magnetism via lattice strain in $\text{Ca}_3\text{Ru}_2\text{O}_7$ and understanding the mechanism is very important.

As this study shows that people can tune the intertwined orders in similar systems by affecting the TMO_6 (TM, transition metal) octahedra's tilting and rotation. I expect this study will inspire more following works in similar materials.

Given the above-mentioned importance, I believe this study is worthy of publication in Nature Communications. As the manuscript is well-written and organized, I recommend its publication without major changes. I suggest the authors move the Appendix to the "Methods" section or the supplemental materials.

We are grateful to the referee for their positive appraisal of our work. We have moved the appendices into a Methods section as advised.

Some other minor comments:

(1) The layered Ruddlesden-Popper ruthenates usually have very large elastic moduli (elastic modulus along the a/b axis could be $>100\text{ GPa}$). I guess this could be the reason that the strain transferring efficiency is low to 20% in Fig. S2(c), as the shear modulus for Stycast 2850FT is $\sim 4\text{ GPa}$ and the method (burying the sample ends in a droplet of the epoxy without Ti plates) would relax the strain quite a lot. The authors should make this clear in the experimental details.

We thank the referee for this insight, which we have included in the experimental details.

(2) Such a study is very challenging and relies on an in-depth collaboration with neutron and synchrotron light sources. I suggest that the authors (especially the authors from ISIS and Diamond light source) could establish the uniaxial strain tuning as a standard sample environment for some experienced users.

The referee will be pleased to hear that the strain setups that we developed at both I16 and WISH are being developed further by the respective beamline/instrument scientists, with the hope that they can become standard user options in the near future.

III. THIRD REFEREE

The article by Dashwood et al. combines a large body of neutron and synchrotron x-ray diffraction work with a theoretical modelling. Both neutron and x-ray experiments were performed under uniaxial strain using 2 different setups from the Razorbill Instruments. The authors chose to study a bilayer ruthenate $\text{Ca}_3\text{Ru}_2\text{O}_7$ with an antiferromagnetic ground state AFM_b , characterized by a propagation vector (001) and an ordered moment along the b-axis of an orthorhombic structure with $Bb2_1m$ space group. At temperatures above ~ 49 K, but below ~ 60 K, the moment is oriented along the a-axis. It is the effect of compressive and tensile strain on this spin reorientation transition that was studied in this work. Recently, a Rashba-based hybridization mechanism was proposed to explain the nature of this transition, Ref. [27] in the manuscript. The interpretation put forward by Dashwood et al. is seemingly at odds with that work. Dashwood et al. claim instead that the transition is driven by a lattice strain.

We thank the referee for their close reading of our manuscript, and hope to convince them of the importance of our work and the care with which it has been carried out.

To support their claim, they show in Fig. 1 results of neutron diffraction at WISH diffractometer. The integrated intensity of magnetic Bragg peaks shows a similar response to both tensile and compressive strain – namely the intensity of the commensurate peak increases with strain whereas the intensity of incommensurate peak decreases. This constitutes the main result of neutron diffraction part. The measurements are reported at 47.7 K, which corresponds to a region of existence of a magnetic cycloid, in which magnetic moments rotate in the ab-plane. Fig. 1b&d do not correctly represent experimental findings – they show AFM_a & AFM_b commensurate antiferromagnetism regions. The outstanding question is what are the changes to the magnetic structure under strain? The authors promised “Unprecedented insight”, but a reader is left guessing about magnetic structure under strain.

The main result of our neutron scattering measurements is simply that we can tune between the AFM_a and AFM_b phases, via the cycloidal phase, using strain. The referee is correct that we did not determine the magnetic structures shown in Fig. 1b and d in this work – they depict the $\text{AFM}_a/\text{AFM}_b$ structures known from previous work (cited in the caption) for illustration purposes. At 47.7 K, these are the phases that are entered under compressive/tensile strain, as corroborated by neutron scattering measurements at other temperatures and our later resonant x-ray scattering results. We did not include a similar diagram of the cycloidal phase because a full depiction of the complex modulated structure would take up significant space, and because the precise structure is not central to our results (we reference two of our previous studies in the manuscript in which the cycloidal phase was discovered [1] and characterised in detail at zero strain [2]). Determining how the cycloidal state itself evolves under strain would require us to measure the intensities of many magnetic peaks, which was not feasible in this first experiment at WISH using the CS200T strain cell.

Further, Fig. 1a shows that a significant part of the sample is not strained, but surely contributes to a measured intensity. How does one resolve response from strained and unstrained parts of a crystal?

The referee is correct that the neutron beam illuminates a large region of the sample, across which there is a significant strain gradient. As described in the Methods, we use cadmium shielding to reduce background from the cell, and this also blocks scattering from the unstrained regions of the sample at the sample plates. The strain gradient across the remaining, illuminated region leads to the finite width of the transitions in Fig. 1c, as we note on page 3 of the manuscript. This, along with our inability to measure the true strain with our neutron setup (see below), motivated us to turn to resonant x-ray scattering to determine the quantitative phase diagrams.

Finally, as authors admit, the cell geometry precludes an access to nuclear Bragg peaks. I salute their honesty, but this indicates that the true strain was not measured in the experiment and the x-axis units in Fig. 1c is only a suggestion.

While we could not determine the true strain in our neutron scattering experiment (as noted on page 3 of the manuscript), we can determine the *applied* strain using the capacitive displacement sensor built into the Razorbill cells (see the Methods). The softness of the epoxy means that not all of this applied strain will be transmitted to the sample (indeed, this is important to prevent any off-axis stresses from breaking the sample) and there will be a difference between the applied and true strains. We note that the majority of previous studies have had to rely on the displacement sensor as their *only* measure of the strain (often using finite element analysis to estimate how much strain is transferred to the sample) [3–7]. As the referee acknowledges below, our subsequent x-ray scattering measurements allow us to determine the true strain experienced by the probed region of the sample.

The authors then proceed with a synchrotron X-ray diffraction work under strain. The setup for this experiment

is markedly different from the one used in the neutron diffraction. Here, the mounting of the sample is asymmetric and only the bottom part of the crystal is glued to moving parts of the cell. This experiment was performed under a compressive strain only, but allowed the authors to quantify the true strain experienced by the lattice, which is an important result. They have also tracked intensities of magnetic Bragg peaks and constructed the temperature-strain magnetic phase diagrams for strain applied along the a - and b -directions. Rather large error bars in Fig. 3a make a direct comparison to the neutron diffraction work difficult. Applied in both directions the strain lowers the transition temperatures to both AFM_a and AFM_b states. It appears that the region of existence of a cycloid (temperature wise) is not affected by the strain. Doesn't it suggest that the modulation is not coupled the strain?

As we explained above, it was not possible to determine how the precise modulation in the ICC phase changes under strain (we did attempt to measure azimuthal dependences in our x-ray experiment, but these were dominated by the changing cross-section of the needle-shaped samples on rotation that could not be subtracted accurately enough to yield reliable results). For this reason, we focused on the transitions between the ICC and commensurate phases, with the direction and slope of the phase boundaries under a - and b -axis strain motivating and constraining our theoretical model.

The theoretical model in Fig. 3c qualitatively reproduces the experimental results, but contains no ICC region.

As the ICC phase is not the focus of this paper, our theoretical work attempted to reproduce the spin-reorientation transition between the AFM_a and AFM_b phases using a minimal model for a perovskite monolayer. As we mention in the theoretical results section of the manuscript, we would expect the modulated phase to appear naturally in our model when the full bilayer is considered, due to an additional uniform Dzyaloshinskii–Moriya interaction that becomes dominant at the transition when the a - and b -axis anisotropies compete.

Crystals of $Ca_3Ru_2O_7$ cleave easily in the ab -plane. Doesn't this create huge strain inhomogeneity with this type of sample mounting, where the bottom part of the sample is under much higher strain than the top one? The authors should comment on this.

The referee is correct that the asymmetric sample mounting on the CS100 cell leads to the strain being transmitted mostly through the lower surface of the sample. As we note on page 3 and in the Methods section of the main text, and discuss in detail in the Supplementary Information, this leads to a slight bending of the sample under strain. The large area detector used in our x-ray scattering measurements enabled us to reconstruct the Bragg peaks in full 3D reciprocal space, and thereby quantify this bending and determine how it affects the strain distribution in the sample. We find that any strain inhomogeneity through the probed region due to this bending is undetectable within our angular resolution.

Ref. 21 reports an observation of ~ 0.1 mm crystalline domains, in which the a - and b -axes are interchanged. The authors should comment how (if) they ensured that their crystals are in mono-domain state. Otherwise the discussion about anisotropic response of a lattice to a strain is redundant.

As-grown crystals of $Ca_3Ru_2O_7$ are indeed typically twinned. As described in the Methods, we identified twin domains using a polarised light microscope and cut our strain samples from single-domain regions. We confirmed the absence of twinning in our strain samples in both the neutron and x-ray scattering experiments.

Some general comments:

A correct identification of the thermodynamic potential has not been not achieved. It would require to work out a clear link between strain-tensor-components, internal degrees-of-freedom, i.e. the tilts and rotations of RuO_6 -octahedra and the electronic system – which is in reality also responsible for the bonding and the moduli of mechanical degrees-of-freedom. Most of the relevant couplings may be non-linear, as the various modes involved (magnetic, electronic/orbital occupation, tilts and rotations vs. strains, i.e. acoustic phonons) have different symmetries. Hence, one has to understand higher-order coupling terms in the thermodynamic potential. Here, all this is executed only implicitly and at a crude level of a tight-binding-model and, hence, the relevance of such coupling as Eq. (C2) appears unclear.

The aim of theoretical aspect of this work was to capture the spin reorientation transition within a minimal effective model, and in doing so shed light onto the transition mechanism, which we believe we have achieved. We also believe that explanation of experimental observations via a minimal, qualitative model that captures the correct phenomenology is valid and has a strong precedent in the field of condensed matter theory. A maximal or first-principles model – that provides quantitative agreement with experimental data – would certainly strengthen our theory, and we hope

that our results motivate such investigations in the future.

Worse, all parameters mentioned seem not to have any units and appear to be meaningless. If the energy units are eV, then for a hopping parameter of $t_0 = 0.33$ eV and $U = 8$ eV seems to be much too large for a reasonable model of the electronic structure in an (almost metallic) ruthenate, although with a stable local moment.

In tight-binding models such as ours, it is the relative sizes of the electronic parameters which are important, not their absolute values. The units may therefore be considered arbitrary. Moreover, the effective hopping parameter is in the range $t \sim 1.00 - 1.25$ in the presence of the self-consistently generated strain, across the temperatures considered. The value of $U/t \sim 8$ is reasonable for a strongly correlated system. An additional axis has been added to Fig. 4 to make clear the magnitude of the effective hopping parameter.

The strains are measured in 'arb. units', but strains are always dimensionless, and should have definite meaning as relative displacement/change of length.

We thank the referee for drawing our attention to this labelling error, which we have amended.

In the Fig. 4 strain for the self-consistent calculation changes by order-of 0.1–0.2 which makes no sense to me. The experiment show strains like 0.005.

We agree that this is a concern and have made some changes to address it. Firstly, we have redefined ε (in Eq. 2 and Fig. 4) such that it is measured with respect to the system at the Néel temperature, which is considered unstrained. This gives meaning to the absolute value of ε . Secondly, our model has a scaling property whereby its solution is invariant under the scaling of the strain parameters as $\nu \rightarrow x\nu$, $\kappa \rightarrow x^2\kappa$. This scales the strain solution as $\varepsilon \rightarrow \varepsilon/x$, but otherwise leaves the solution of the model (i.e. the phase diagrams) unchanged. In our revision, we have chosen a scaling that places the theoretical values of the change in strain across the SRT within reasonable agreement with the observed values ($\Delta\varepsilon \sim 0.0015$ compared to $\Delta\varepsilon \sim 0.0005$). Moreover, the strain-hopping coupling strength, ν , is comparable in magnitude to a value for Sr_2RuO_4 that has been calculated from first-principles [8]. The data in Fig. 3c and 4 have been recalculated using these scaled parameters.

If the model has any meaning, the authors should demonstrate that it can reproduce at least a correct order-of-magnitude for basic mechanical properties (bulk modulus, thermal expansion, elastic constants).

We believe that our minimal model does a good job of capturing a non-trivial interplay of electronic and lattice effects. This interplay is one of a number of contributions to the bulk mechanical properties and aside from the tilts and rotations – which are incorporated into our theory and do occur with the correct order of magnitude – may be a subdominant effect on bulk mechanical properties.

Therefore, based on such a proposal and a schematic model, the authors cannot claim to have identified the correct mechanism. For that, they should make a much more quantitative effort specific to the material. Hence, this is a proposal to rationalize their experimental data, but no resolution of the issue.

We respectfully disagree with the referee. Both *ab initio* and minimal models have their role to play. Our model captures a subtle interplay of lattice and electronic structure whose drivers would be considerably harder to identify in a calculation that included the all potential microscopic couplings. A complete *ab initio* calculation and quantitative comparison would be useful for future work but is beyond the scope of this work.

-
- [1] C. D. Dashwood, L. S. I. Veiga, Q. Faure, J. G. Vale, D. G. Porter, S. P. Collins, P. Manuel, D. D. Khalyavin, F. Orlandi, R. S. Perry, R. D. Johnson, and D. F. McMorrow, *Phys. Rev. B* **102**, 180410(R) (2020).
 - [2] Q. Faure, C. D. Dashwood, C. V. Colin, R. D. Johnson, E. Ressouche, G. B. G. Stenning, J. Spratt, D. F. McMorrow, and R. S. Perry, *Phys. Rev. Res.* **5**, 013040 (2023).
 - [3] C. W. Hicks, D. O. Brodsky, E. A. Yelland, A. S. Gibbs, J. A. N. Bruin, M. E. Barber, S. D. Edkins, K. Nishimura, S. Yonezawa, Y. Maeno, and A. P. Mackenzie, *Science* **344**, 283 (2014).
 - [4] A. Steppke, L. Zhao, M. E. Barber, T. Scaffidi, F. Jerzembeck, H. Rosner, A. S. Gibbs, Y. Maeno, S. H. Simon, A. P. Mackenzie, and C. W. Hicks, *Science* **355**, eaaf9398 (2017).

- [5] P. Malinowski, Q. Jiang, J. J. Sanchez, J. Mutch, Z. Liu, P. Went, J. Liu, P. J. Ryan, J.-W. Kim, and J.-H. Chu, Nat. Phys. **16**, 1189 (2020).
- [6] H.-H. Kim, E. Lefrançois, K. Kummer, R. Fumagalli, N. B. Brookes, D. Betto, S. Nakata, M. Tortora, J. Porras, T. Loew, M. E. Barber, L. Braicovich, A. P. Mackenzie, C. W. Hicks, B. Keimer, M. Minola, and M. Le Tacon, Phys. Rev. Lett. **126**, 037002 (2021).
- [7] T. Worasaran, M. S. Ikeda, J. C. Palmstrom, J. A. W. Straquadine, S. A. Kivelson, and I. R. Fisher, Science **372**, 973 (2021).
- [8] Y.-S. Li, M. Garst, J. Schmalian, S. Ghosh, N. Kikugawa, D. A. Sokolov, C. W. Hicks, F. Jerzembeck, M. S. Ikeda, Z. Hu, B. J. Ramshaw, A. W. Rost, M. Nicklas, and A. P. Mackenzie, Nature **607**, 276 (2022).

Reviewers' Comments:

Reviewer #1:

Remarks to the Author:

The authors have revised their manuscript satisfactorily. I therefore recommend publication in Nature Communications.

Reviewer #2:

Remarks to the Author:

The authors have addressed my concerns and revised the manuscript accordingly. I recommend its publication in Nature communications.

Reviewer #3:

Remarks to the Author:

The authors mended some of the simpler deficiencies of the model, but as it is now, I still do not think the approach is able to support the relatively far reaching conclusions about the identification of the correct mechanism for the SRT and also metal-insulator transition(s) in $\text{Ca}_3\text{Ru}_2\text{O}_7$.

A qualitative or semi-quantitative model is fair enough, if the parameters used for the illustration of a mechanism are in some reasonable correspondence to the materials where effects have been seen. A simple model should bear at least some superficial resemblance to the underlying electronic structure. This is not a demand to provide a fully quantitative evaluation or simulation of the material.

So I will continue from the point where I stopped in the 1st round of review. The authors still use $U/t \sim 8$ for the electronic structure of $\text{Ca}_3\text{Ru}_2\text{O}_7$ and claim that Ru is a strongly correlated material.

This is not tenable, the material is clearly in an intermediate state and, depending on the regime, close or really metallic. If the simple model really requires such strong correlation effects to reproduce the qualitative behavior, then the approach appears questionable, at best.

Too large U , in their ansatz for the free energy seems to provide a large lever for the modification of the lattice to tune the magnetic state and the on-site electronic occupation. Therefore, the magnitude of this term is critical.

The problem may be rooted in the simultaneous use of U for the electronic part as a microscopic Hubbard-like correction, and as parameter in the free energy Eq.(2) which is written like a double-counting correction in around-mean-field ansatz used in DFT approaches. The mean-field Hartree-Fock evaluation of the electronic problem may justify such an ansatz. The insertion of this term as the leading contribution to the free energy expansion from charge and magnetic moments as order-parameters at finite temperatures is not an obvious choice, this parameter should be a result of solving the statistical physics problem on top of solving for the electronic structure. The authors do not provide a discussion regarding this. As such this approach would require at least some justification to support validity of this ansatz and help to understand the role of the coupling parameters.

The authors did not take up the offer to re-evaluate their "microscopic" model in the light of a straightforward phenomenological approach to the required spin-orbit-lattice-couplings too.

Therefore, the changes and responses do not seem satisfactory.

Overall I do not think the model, as developed and evaluated in this manuscript, is able to support the conclusions about identification of the correct mechanism for the metal-insulator-transition and SRT in $\text{Ca}_3\text{Ru}_2\text{O}_7$. If the authors wish to publish the experimental results along with this model, they should considerably reduce the strength of their conclusions in that respect and present it as a sketchy proposal to rationalize their findings - not as a solution of the problem.

I am also not convinced that this approach will be very useful to analyse any other material with similar magneto-elastic-electronic coupling phenomena, unless some link to more realistic representation of the electronic structure and its coupling to the magnetic degrees of freedom etc can be formulated and employed, at least at a qualitatively realistic level.

Reply to Referees for “Strain control of a bandwidth-driven spin reorientation in $\text{Ca}_3\text{Ru}_2\text{O}_7$ ”

I. FIRST REFEREE

The authors have revised their manuscript satisfactorily. I therefore recommend publication in Nature Communications.

We thank the referee for their review of our manuscript and for their positive recommendation.

II. SECOND REFEREE

The authors have addressed my concerns and revised the manuscript accordingly. I recommend its publication in Nature communications.

We thank the referee for their review of our manuscript and for their positive recommendation.

III. THIRD REFEREE

The authors mended some of the simpler deficiencies of the model, but as it is now, I still do not think the approach is able to support the relatively far reaching conclusions about the identification of the correct mechanism for the SRT and also metal-insulator transition(s) in $\text{Ca}_3\text{Ru}_2\text{O}_7$.

We thank the referee for their further comments, and hope to convince them of the suitability of our approach for the task at hand.

A qualitative or semi-quantitative model is fair enough, if the parameters used for the illustration of a mechanism are in some reasonable correspondence to the materials where effects have been seen. A simple model should bear at least some superficial resemblance to the underlying electronic structure. This is not a demand to provide a fully quantitative evaluation or simulation of the material.

So I will continue from the point where I stopped in the 1st round of review. The authors still use $U/t \sim 8$ for the electronic structure of $\text{Ca}_3\text{Ru}_2\text{O}_7$ and claim that Ru is a strongly correlated material.

This is not tenable, the material is clearly in an intermediate state and, depending on the regime, close or really metallic. If the simple model really requires such strong correlation effects to reproduce the qualitative behavior, then the approach appears questionable, at best.

We respectfully disagree with the referee on this point. A value $U/t \sim 8$ is consistent with both first-principles theoretical and experimental studies of $\text{Ca}_3\text{Ru}_2\text{O}_7$, as per the following evidence.

Firstly, the tight binding model that we employ in our theoretical model has been fitted to ARPES data, and the value of the nearest-neighbour hopping parameter that best fits the electronic structure is $t \approx 0.12$ eV [1]. Moreover, the value of screened Coulomb repulsion, U , has been estimated for this system by fitting the results of DFT+ U calculations to experiment, and the fermiology of the material is best captured by a value of $U \approx 1.6$ eV [2]. These values suggest that a value of $U/t \sim 8$ is entirely reasonable for the system, and places $\text{Ca}_3\text{Ru}_2\text{O}_7$ as a moderately-to-strongly correlated metal (our self-consistent calculations actually give a range of $6.4 \leq U/t \leq 7.7$, and a value of $U/t \approx 6.6$ at the spin reorientation transition, as can be seen in Fig. 4 of the manuscript). We have added statements to the Introduction and the caption of Fig. 4 in the manuscript to clarify this point.

This characterisation of $\text{Ca}_3\text{Ru}_2\text{O}_7$ as a moderately-to-strongly correlated metal is further supported by the fact that it is close to a Mott insulating phase, which can be accessed with small amounts of Ti doping [3]. The doped system has also been studied from first-principles using both DFT+ U and DFT+DMFT approaches, using a Hubbard interaction of $U \approx 3$ eV to 4 eV on the Ruthenium ions, which successfully captures the phenomenology of the system

[4]. This again supports a relatively large value of U/t in this material.

It is also worth mentioning that strong screened interactions are understood to feature across the family of ruthenates. In Mott-insulating Ca_2RuO_4 , for example, it is generally agreed that hopping and Hubbard parameter values of $t \approx 0.2\text{ eV}$ and $U \approx 3\text{ eV}$ to 4 eV accurately represent the system [5–7]. A compelling case has been made that the surprisingly strong intra-orbital electronic correlations in the ruthenates is a result of their enhancement by Hund’s coupling [8, 9] – making these materials so-called ‘Hund’s metals’. We do not explicitly include the Hund’s coupling in our electronic model since we found that its inclusion has no effect on the magnetic solution, but instead assume that its leading-order effect is to enhance the value of U .

Strong correlations do not preclude a metallic state, therefore, and that there is strong evidence that $\text{Ca}_3\text{Ru}_2\text{O}_7$ is indeed a moderately-to-strongly correlated metal.

Too large U , in their ansatz for the free energy seems to provide a large lever for the modification of the lattice to tune the magnetic state and the on-site electronic occupation. Therefore, the magnitude of this term is critical.

Although U is a parameter of the model that is set by hand, we would like to emphasise that the value of t is determined self-consistently through its coupling to the lattice strain. The ratio of U/t – which is the quantity of importance – is therefore also determined self-consistently. Moreover, the value of $U/t \approx 6.6$ that we find is consistent with *ab initio* values that have been fitted to experiment, as discussed above.

The problem may be rooted in the simultaneous use of U for the electronic part as a microscopic Hubbard-like correction, and as parameter in the free energy Eq. (2) which is written like a double-counting correction in around-mean-field ansatz used in DFT approaches. The mean-field Hartree-Fock evaluation of the electronic problem may justify such an ansatz. The insertion of this term as the leading contribution to the free energy expansion from charge and magnetic moments as order-parameters at finite temperatures is not an obvious choice, this parameter should be a result of solving the statistical physics problem on top of solving for the electronic structure. The authors do not provide a discussion regarding this. As such this approach would require at least some justification to support validity of this ansatz and help to understand the role of the coupling parameters.

The free energy of our model is, in fact, determined by solving the electronic problem using statistical techniques, as follows. First, a mean-field decoupling of the Hubbard interaction term in the electronic Hamiltonian is performed, in both the charge and spin channels. Then, the free energy of the grand canonical ensemble is evaluated *via* the partition function, to give Eq. (2) (in the absence of strain). This result is the Hartree-Fock free energy of our electronic model, when considered at finite temperature. It should be noted that the Hubbard term in the free energy arises from the mean-field decoupling of the Hubbard interaction, and is not introduced by hand. We have added a few sentences in the Methods in order to clarify this procedure. Finally, we wish to emphasise that this is a method of treating charge/magnetic order in Hubbard models, within a mean-field approximation, that has a strong precedent.

The authors did not take up the offer to re-evaluate their “microscopic” model in the light of a straightforward phenomenological approach to the required spin-orbit-lattice-couplings too. Therefore, the changes and responses do not seem satisfactory.

Overall I do not think the model, as developed and evaluated in this manuscript, is able to support the conclusions about identification of the correct mechanism for the metal–insulator-transition and SRT in $\text{Ca}_3\text{Ru}_2\text{O}_7$. If the authors wish to publish the experimental results along with this model, they should considerably reduce the strength of their conclusions in that respect and present it as a sketchy proposal to rationalize their findings – not as a solution of the problem.

When taken together, we believe that our experimental data and theoretical model present a strong case for a strain-driven mechanism behind the SRT in $\text{Ca}_3\text{Ru}_2\text{O}_7$. We would like to note that it is not just the agreement with our experimental applied-strain data that gives us confidence in our theoretical results, but also the agreement with our temperature-dependent neutron diffraction data (with no applied strain). This data corroborates the increase in internal strain and nearest-neighbour hopping on cooling through the transition, as predicted by our model, which occur due to changes in the tilts and rotations of the RuO_6 octahedra.

I am also not convinced that this approach will be very useful to analyse any other material with similar magneto-elastic-electronic coupling phenomena, unless some link to more realistic representation of the electronic structure and

its coupling to the magnetic degrees of freedom etc can be formulated and employed, at least at a qualitatively realistic level.

As noted above, the features that we identify as giving rise to the SRT (moderate-to-strong correlations, the importance of octahedral tilts/rotations, etc.) are not unique to $\text{Ca}_3\text{Ru}_2\text{O}_7$, but are shared by a range of other transition-metal oxides – especially the other Ruddlesden-Popper ruthenates. We are therefore hopeful that our results will inspire further investigations of the role of strain in these topical materials.

-
- [1] M. Horio, Q. Wang, V. Granata, K. P. Kramer, Y. Sassa, S. Jöhr, D. Sutter, A. Bold, L. Das, Y. Xu, R. Frison, R. Fittipaldi, T. K. Kim, C. Cacho, J. E. Rault, P. L. Fèvre, F. Bertran, N. C. Plumb, M. Shi, A. Vecchione, M. H. Fischer, and J. Chang, npj Quantum Materials **6**, 29 (2021).
 - [2] D. Puggioni, M. Horio, J. Chang, and J. M. Rondinelli, Phys. Rev. Res. **2**, 023141 (2020).
 - [3] X. Ke, J. Peng, D. J. Singh, T. Hong, W. Tian, C. R. Dela Cruz, and Z. Q. Mao, Phys. Rev. B **84**, 201102 (2011).
 - [4] F. Lechermann, Q. Han, and A. J. Millis, Phys. Rev. Res. **2**, 033490 (2020).
 - [5] E. Gorelov, M. Karolak, T. O. Wehling, F. Lechermann, A. I. Lichtenstein, and E. Pavarini, Phys. Rev. Lett. **104**, 226401 (2010).
 - [6] S. Mohapatra and A. Singh, J. Phys. Condens. Matter **32**, 485805 (2020).
 - [7] G. Khaliullin, Phys. Rev. Lett. **111**, 197201 (2013).
 - [8] J. Mravlje, M. Aichhorn, T. Miyake, K. Haule, G. Kotliar, and A. Georges, Phys. Rev. Lett. **106**, 096401 (2011).
 - [9] A. Georges, L. de' Medici, and J. Mravlje, Annu. Rev. Condens. Matter Phys. **4**, 137 (2013).

Reviewers' Comments:

Reviewer #3:

Remarks to the Author:

The justification of the model now appears satisfactory -
at least at a qualitative or semiquantitative level of this model,
the experimental part is strong enough, I recommend publishing the paper.

Reply to Referees for “Strain control of a bandwidth-driven spin reorientation in $\text{Ca}_3\text{Ru}_2\text{O}_7$ ”

I. THIRD REFEREE

The justification of the model now appears satisfactory – at least at a qualitative or semiquantitative level of this model, the experimental part is strong enough, I recommend publishing the paper.

We thank the referee for their positive recommendation, and for helping us to refine our manuscript.